:ᐧᶜ: PLOS | ONE

# Genetic polymorphism in C3 is associated with progression in chronic kidney disease (CKD) patients with IgA nephropathy but not in other causes of CKD

**Sara T. Ibrahim**[1,2,3‡]*, **Rajkumar Chinnadurai**[2,3☉], **Ibrahim Ali**[2,3☉], **Debbie Payne**[3], **Gillian I. Rice**[4], **William G. Newman**[5‡], **Eman Algohary**[1‡], **Ahmed G. Adam**[1‡], **Philip A. Kalra**[2,3☉]

1 Department of Internal Medicine and Nephrology, Faculty of Medicine, Alexandria University, Alexandria, Egypt, 2 Department of Renal Medicine, Salford Royal NHS Foundation Trust, Salford, United Kingdom, 3 University of Manchester, Manchester, United Kingdom, 4 Manchester Centre for Genomic Medicine, Manchester University NHS Foundation Trust, Health Innovation Manchester, Manchester, United Kingdom, 5 Division of Evolution and Genomic Sciences, Faculty of Biology, Medicine and Human Sciences, University of Manchester, Manchester, United Kingdom

☉ These authors contributed equally to this work.
‡ These authors also contributed equally to this work.
* sara.ibrahim@alexmed.edu.eg

**Data Availability Statement:** All relevant data are within the paper and its Supporting Information files.

## Abstract

### Objectives

The *R102G* variant in complement 3 (*C3*) results in two allotypic variants: *C3 fast* (*C3F*) and *C3 slow* (*C3S*). *C3F* presents at increased frequency in patients with chronic kidney disease (CKD), our aim is to explore its role in CKD progression and mortality.

### Methods

Delta (Δ) eGFR for 2038 patients in the Salford Kidney Study (SKS) was calculated by linear regression; those with $\leq$-3ml/min/1.73m$^2$/yr were defined as rapid progressors (RP) and those with ΔeGFR between -0.5 and +1ml/min/1.73m$^2$/yr, labelled stable CKD patients (SP).A group of 454 volunteers was used as a control group. In addition, all biopsy-proven glomerulonephritis (GN) patients were studied regardless of their ΔeGFR. *R102G* was analysed by real-time PCR, and genotypic and allelic frequencies were compared between RP and SP along with the healthy control group.

### Results

There were 255 SP and 259 RP in the final cohort. Median ΔeGFR was 0.07 vs. -4.7 ml/min/1.73m$^2$/yr in SP vs. RP. *C3F* allele frequency was found to be significantly higher in our CKD cohort (25.7%) compared with the healthy control group (20.6%); p = 0.008.However, there was no significant difference in *C3F* allele frequency between the RP and SP groups. In a subgroup analysis of 37 patients with IgA nephropathy in the CKD cohort (21 RP and 16 SP), there was a significantly higher frequency of *C3F* in RP 40.5% vs. 9.4% in SP; p =

**Funding:** The author(s) received no specific funding for this work.

**Competing interests:** The authors have declared that no competing interests exist.

0.003. In the GN group, Cox regression showed an association between *C3F* and progression only in those with IgA nephropathy (n = 114);HR = 1.9 (95% CI 1.1–3.1; p = 0.018) for individuals heterozygous for the *C3F* variant, increased further for individuals homozygous for the variant, HR = 2.8 (95% CI 1.2–6.2; p = 0.014).

## Conclusion

The *C3 variant R102G* is associated with progression of CKD in patients with IgA nephropathy.

## Introduction

Chronic kidney disease (CKD) is a worldwide health concern due to the high morbidity and mortality associated with CKD progression and end stage renal disease (ESRD) [1,2]. Multiple risk factors play a role in CKD progression [3,4]. Better predictive tools to characterise patients'future risk of progression could help in delivering targeted treatment to high-risk patients. Therefore, there has been increased attention in elucidating the genetic determinants of CKD progression given patients with the same phenotypic risk profile progress at different rates.

Several genome-wide association studies (GWAS) identified numerous genetic loci associated with CKD [5–8]. Unfortunately, the lack of clinical and biochemical characteristics of the populations involved in these large GWAS and the absence of longitudinal data makes it difficult to determine if these genetic loci are associated with CKD progression or specific causes of CKD. These important issues can be addressed by targeted studies considering a limited number of genetic polymorphisms, but in a well-known CKD cohort with sufficient period of follow-up [9,10].

Although the complement system has an important role in our innate immune system, it is involved in the pathogenesis of most glomerular and tubular kidney diseases [11].The third human complement component, C3, plays a pivotal role in the complement system cascade, such that a polymorphism in its genetic coding can affect the activity of the complement system resulting in more activation, inflammation and tissue destruction [12,13].An extensively studied variant in*C3* is *R102G* (rs2230199), with three polymorphic variants: homozygote *C3SS* (slow), homozygote *C3FF* (fast), and heterozygote *C3FS*. The fast and slow description for C3 activity was known even before the discovery of this single nucleotide polymorphism (SNP); the fast or slow labelling refers to the speed of movement of C3 on the agarose gel during protein electrophoresis [14].The *R102G* polymorphism is determined by a change in one nucleotide where cytosine is replaced by guanine at position 364 in the exon 3 of the β chain in the *C3* gene on chromosome 19.This results in the replacement of a positively charged arginine amino acid in the *C3S* allele by a neutral glycine amino acid in the *C3F* allele [15]. The neutral glycine amino acid in the *C3F* allele decreases its ability to bind to complement factor H, an important complement regulator protein, leading to less regulation and more activation of the complement system [12].The frequency of the *C3F* allele differs in different ethnicities; it is 20% in Caucasian, 5% in blacks and 1% in Asian populations [16].

The *C3F* variant has been associated with an increased risk for age related macular degeneration [17],rheumatoid arthritis and Crohn's disease[18].In kidney diseases, *C3F* has been found to be more prevalent in CKD[19] especially in glomerulonephritis (GN), such as; membranoproliferative GN type II (MPGN II) [20], IgA nephropathy (IgAN)[21] and systemic vasculitis[22].Hence we have designed our study, aiming to investigate the association between

the *C3 variant R102G* and CKD progression and all-cause mortality in all-cause CKD, as well as in different specific CKD causes, in a large, well characterised non-dialysis CKD cohort (stages 3–5) with a long period of follow-up.

## Materials and methods

### Study population

Patients were ascertained through the Salford Kidney Study (SKS), a large non-dialysis CKD cohort that has recruited and followed-up CKD patients referred to Salford renal service prospectively since 2002 [23–25].The participants are ≥ 18 years with estimated glomerular filtration rate (eGFR) <60 ml/min/1.73m$^2$ and have not started renal replacement therapy (RRT). Data recorded at baseline includes demographics (age, sex, weight, height, CKD aetiology, smoking status, alcohol history and functional status), comorbid conditions, medications and laboratory results. Blood samples including whole blood, serum and plasma are collected and stored at -80˚C for future research, with most recruited patients having extracted DNA that is also stored. All patients are followed up annually until reaching an SKS endpoint which includes: a) RRT commencement b) death c) lost to follow-up or discharge from clinic and d) unable to participate, or withdrawal of consent. The study complies with the declaration of Helsinki and ethical approval has been obtained from the regional ethical committee (current REC reference 15/NW/0818).

Study 1 included 514 CKD patients (255 stable CKD patients and 259 rapid progressors) from patients recruited into the Salford kidney study (SKS) and 454 ethnically matched individuals from a bank of anonymized, healthy, unrelated individuals through the regional molecular genetics laboratory of Manchester Royal Infirmary hospital.

As most of the literature has emphasized the role of the complement system in GN[26] and because an increase of the C3F allele frequency has been found in various GN types[20–22], we designed the second part of the study in order to increase the number of GN patients.

Hence, study 2 included 269 GN patients recruited into the SKS.

**Patient selection.** Study 1:the total number of patients who were recruited into SKS from October 2002 to December 2016 was 3115.The eGFR was calculated, for patients who had a complete data set, from creatinine values obtained during outpatient's visits by the CKD Epidemiology Collaboration (CKD-EPI) formula[27]. To ensure an adequate follow-up period to enable determination of the rate of CKD progression, all patients with <2 years follow-up or <4 available eGFR values were excluded. The delta (Δ) eGFR (±ml/min/1.73m$^2$/yr) was calculated for remaining patients by linear regression. We defined rapid progressors (RP) as patients with ΔeGFR ≤-3ml/min/1.73m$^2$/yr and stable CKD patients (SP)as patients with ΔeGFR between -0.5 and +1ml/min/1.73m$^2$/yr. As it is well-known that acute kidney injury (AKI) episodes can leave some irreversible effects and can contribute to CKD progression [28,29], it was also important to exclude the patients who had AKI during their follow-up period. The resultant cohort of RP and SP was further refined by two independent researchers reviewing patients'eGFR-time graphs in the hospital electronic records such that those with AKI episodes contribute to the fast decline in eGFR or treatment effects contribute to improving or stability of eGFR were excluded. The final cohort was then derived from the above patients who had available DNA samples for analysis (Fig 1).

Study 2: All biopsy proven GN patients in the SKS, regardless of their ΔeGFR, and with available DNA samples totalled 269 GN patients. These included 114 IgAN, 50 focal segmental glomerulosclerosis (FSGS), 59 membranous nephropathy, and 46 other types of GN including minimal change disease, post streptococcus GN, MPGN. (Fig 2)

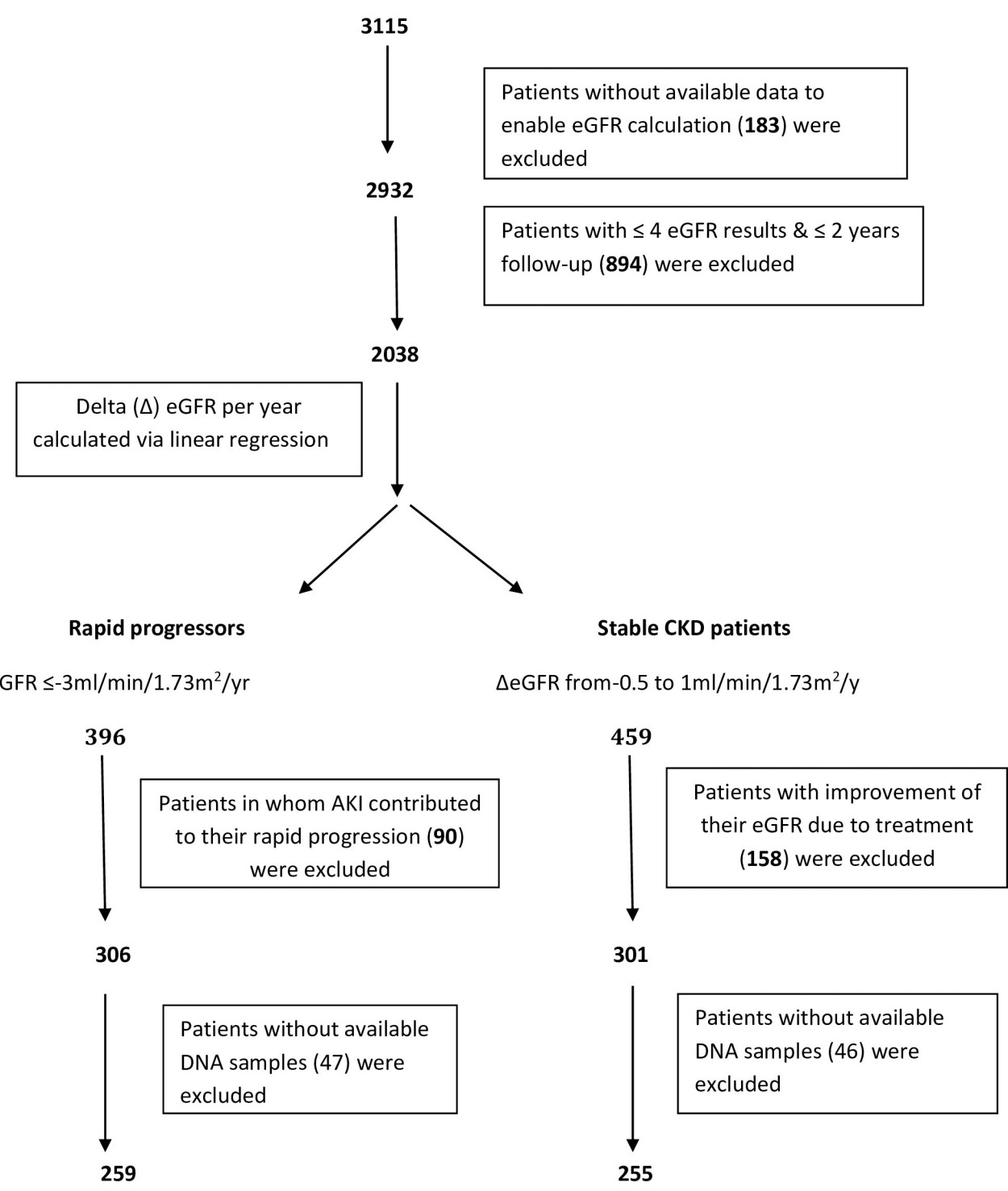

**Fig 1. Flowchart of patient recruitment to study 1.**

Total patients in SKS who were recruited from October 2002 to December 2016

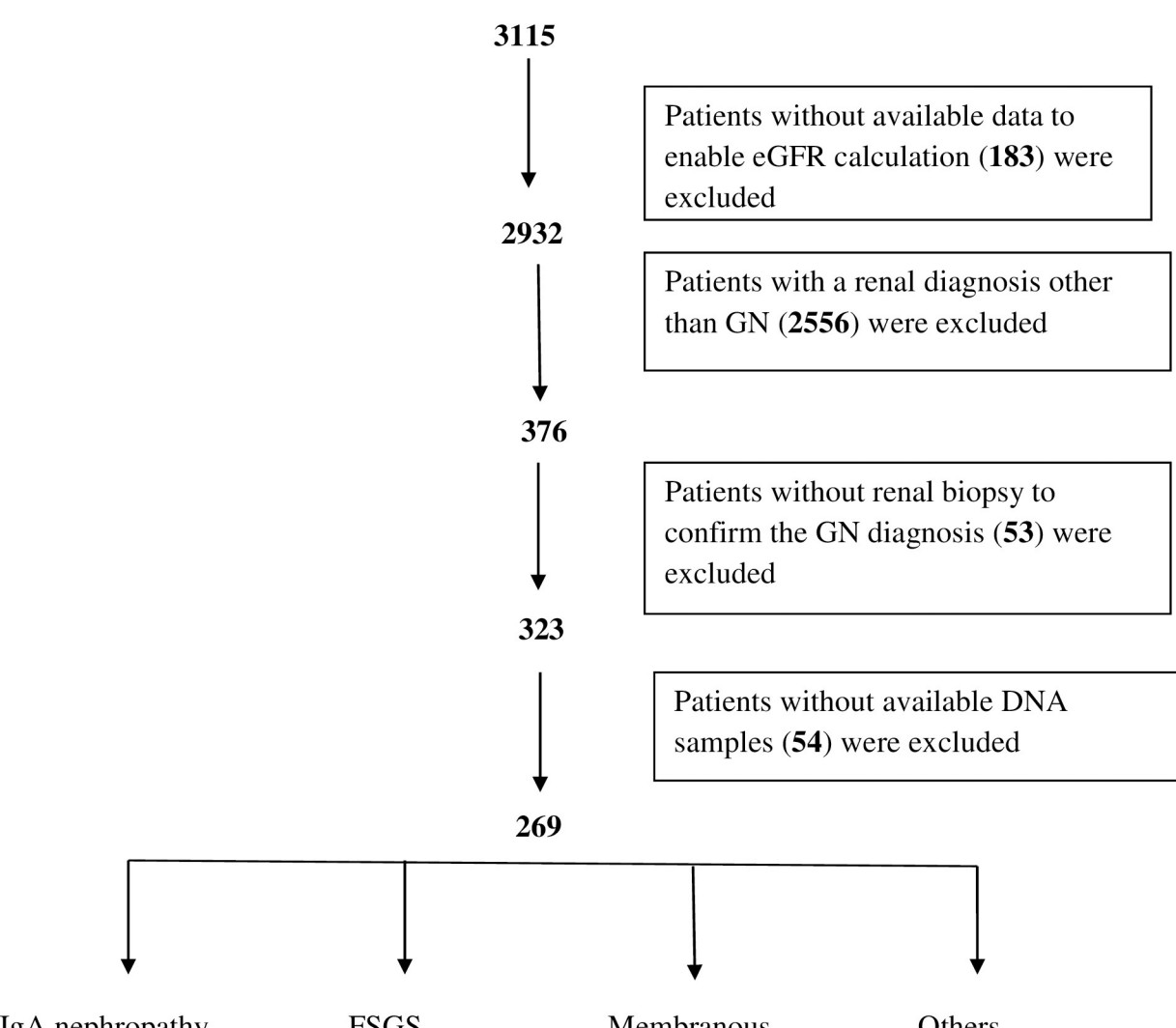

**Fig 2. Flowchart of patient recruitment to study 2.**

## Data gathering and study outcomes

Demographic data and laboratory results were collected from electronic patient records (EPR) and the SKS data base at baseline. Two main outcomes were studied:

1. CKD progression: As competing risks can lead to inaccurate survival analysis with Cox regression, we used composite end point [30]. CKD progression was defined as ESRD (reaching RRT or eGFR<10 ml/min/1.73m2) or ≥40% eGFR decline. This is a validated end point for CKD progression used in many trials [31–33].

2. All-cause mortality.

## Genotyping of the *R102G SNP*

The participants were genotyped for the *R102G C3 polymorphism* (*rs2230199*) by a validated TaqMan SNP Genotyping Assay. SNP genotyping was performed by the Applied Biosystem Step One™ Real-Time PCR system according to the manufacturer's recommendations (Thermo Fisher Scientific). The assay mix (including unlabelled PCR primers, and FAM and VIC dye-labelled TaqMan MGB probes) was designed by Thermo Fisher Scientific. The reaction system utilised 20 ng of genomic DNA, 5 $\mu$l of TaqMan Universal PCR Master Mix, and 0.5 $\mu$l 20× Assay Mix and was adjusted with water for a total volume of 10 $\mu$l in each well. Alleles were scored using Applied Biosystem Step One™ Real-Time PCR software (version 2.1).

## Statistical analysis

Continuous data are expressed as the median and interquartile range (IQR), and categorical data as frequency and percentage. Comparisons between groups were undertaken with the Mann-Whitney U test, chi-squared test,Monte-Carlo test, or Fisher-Exact test as appropriate. The relationship between the *C3 SNP* and the incidence rate of renal events or death was investigated by univariate and multivariate Cox regression analyses and Kaplan–Meier survival curves. All risk factors for renal progression including: age, gender, smoking, diabetes, hypertension, albumin, haemoglobin, urinary protein:creatinine ratio, and baseline eGFR were tested in the univariate and multivariate Cox regression. The same factors were included in the univariate model for mortality with the addition of other risk factors including history of cancer, myocardial infarction, congestive cardiac failure (CCF) and C-reactive protein level (CRP) and those variables that showed significant association in the univariate model were included in the multivariate one. In the second part of the study, when the relationship between the *C3 SNP* and renal progression in different GN groups was investigated, active treatment with corticosteroids or immunosuppressive therapies was included in the univariate and multivariate Cox regression analyses.A p-value <0.05 was considered statistically significant throughout the analyses. All statistical analyses were performed using SPSS(Version 23) (IBM SPSS, Chicago, IL) licensed to the University of Manchester.

## Results

### Study 1 (CKD cohort)

**Baseline characteristics.** Baseline characteristics of the entire CKD cohort and comparison between RP and SP patients are summarized in Tables 1 and 2.

The median age of the total cohort was 62.8 years(IQR50.5–73.7) with more males (62.2%) and Caucasian ethnicity (96%).The RP patients were significantly younger in age (median 56; IQR45.6–69.3 years) than SP (median68.4; IQR 57.4–76.5 years); p-value <0.001 and included fewer males (52.1%) than SP (73.3%). Diabetes mellitus was the commonest cause of CKD in the total cohort (19.6%), while membranous nephropathy was the least (2.3%). Autosomal dominant polycystic kidney disease constituted the commonest cause in the RP group (21.2%). Median ΔeGFR was 0.07; (IQR -0.25 to 0.49)ml/min/1.73m$^2$/yrin SP vs. -4.7(IQR-6.4 to -3.7)ml/min/1.73m$^2$/yrin RP. Baseline eGFR was significantly higher in RP (median 31.4, IQR 23.1–41.4ml/min/1.73m$^2$/yr) than that in SP (median 24.7, IQR 18.2–33.1ml/min/1.73m$^2$/yr).

**The *R102G C3* polymorphism (*rs2230199*).** The distribution of the polymorphism frequencies in both the CKD cohort and the controls were consistent with the Hardy-Weinberg equation (S1 Table). There were significant differences between the CKD group in genotypic

**Table 1. Baseline clinical characteristics of the CKD cohort and comparison between rapid progressors and stable CKD patients.**

| | Total CKD (n = 514) | Stable patients (n = 255) | Rapid progressors (n = 259) | p-Value |
|---|---|---|---|---|
| Age (years) | 62.8(50.5–73.7) | 68.4 (57.4–76.5) | 56 (45.6–69.3) | <**0.001** |
| Gender (male), n (%) | 322 (62.6%) | 187 (73.3%) | 135 (52.1%) | <**0.001** |
| Ethnicity (Caucasian), n (%) | 494 (96%) | 250 (98%) | 244 (94.2%) | **0.025** |
| Smoking, n (%) | 323 (62.8%) | 161 (63.1%) | 162 (62.5%) | 0.89 |
| HTN, n (%) | 495 (96.3%) | 245 (96.1%) | 250 (96.5%) | 0.78 |
| Systolic BP (mmHg) | 138 (125–150) | 136 (124–150) | 138 (127–152) | 0.11 |
| Diastolic BP (mmHg) | 76 (70–82) | 75 (66–80) | 78 (70–84) | <**0.001** |
| DM, n (%) | 158 (30.7%) | 91 (35.7%) | 67 (25.9%) | **0.016** |
| MI, n (%) | 46 (8.9%) | 33 (12.9%) | 13 (5%) | **0.002** |
| CCF, n (%) | 21 (4%) | 11 (4.3%) | 10 (3.9%) | 0.80 |
| PVD, n (%) | 32 (6.2%) | 19 (7.5%) | 13 (5%) | 0.25 |
| CVA, n (%) | 20 (3.9%) | 9 (3.5%) | 11 (4.2%) | 0.67 |
| Tumor, n (%) | 52 (10.1%) | 34 (13.4%) | 18 (6.9%) | **0.016** |
| CKD cause, n (%) | | | | |
| DM | 101 (19.6%) | 51 (20%) | 50 (19.3%) | 0.84 |
| HTN | 55 (10.7%) | 34 (13.3%) | 21 (8.1%) | 0.06 |
| RVD | 32 (6.2%) | 20 (7.8%) | 12 (4.6%) | 0.13 |
| IgA nephropathy | 37 (7.2%) | 16 (6.3%) | 21 (8.1%) | 0.34 |
| FSGS | 16 (3.1%) | 5 (2%) | 11 (4.2%) | 0.14 |
| Membranous GN | 12 (2.3%) | 5 (2%) | 7 (2.7%) | 0.58 |
| Other GN &vasculitis | 33 (6.4%) | 17 (6.7%) | 16 (6.2%) | 0.68 |
| ADPKD | 57 (11.1%) | 2 (0.8%) | 55 (21.2%) | <**0.001***|
| Pyelonephritis and interstitial nephritis | 67 (13%) | 42 (16.5%) | 25 (9.7%) | **0.021** |
| Unknown | 60 (11.6%) | 38 (14.9%) | 22 (8.5%) | **0.023** |
| Others | 44 (8.5%) | 25 (9.8%) | 19 (7.3%) | 0.28 |

HTN-hypertension, BP-blood pressure, mmHg-millimeter of mercury, DM-diabetes mellitus, MI-myocardial infarction, CCF-congestive cardiac failure, PVD-peripheral vascular disease, CVA-cerebrovascular accident, CKD-chronic kidney disease, RVD-renovascular disease, FSGS- focal segmental glomerular sclerosis, GN-glomerulonephritis, ADPKD-autosomal dominant polycystic kidney disease.

Continuous variables are expressed as median (interquartile range) and p-Value by Man-Whitney U test.

Categorical variables are expressed as number (%) and p-Value by Chi-Square test.

*P-Value by Fisher-Exact test.

frequencies(*FF* 9.3%, *FS* 32.7%, *SS* 58%) and allele frequencies (*F* 25.7%, *S* 74.3%) compared with the control group which had genotype variants (*FF* 5.7%, *FS* 29.7%, *SS* 64.6%) and allele frequencies(*F* 20.6%, *S* 79.4%), with p-value = 0.039 and 0.008, respectively (Table 3). Comparison of clinical and biochemical characteristics between CKD patients with the rare genotype (*C3FF*) and the commoner genotypes (*C3FS* and *C3SS*) is summarized in S2 Table.

The difference between the RP genotype variants (*FF* 11.2%, *FS* 31.7%, *SS* 57.1%) and allele frequencies (*F* 27%, *S* 73%), and the SP genotype variants (*FF* 7.5%, *FS* 33.7%, *SS* 58.8%) and allele frequencies (*F* 24.3%, *S* 75.7%) was not significant with p-value 0.34 and 0.32, respectively (Table 3). However, in the subgroup analysis of patients stratified by CKD cause, the IgAN group showed a significant difference between RP (n = 21) allele frequencies (*F* 40.5%, *S* 59.5%), and SP (n = 16) allele frequencies (*F* 9.4%, *S* 90.6%), p-value = 0.003, with odds ratio = 6.6 (S3 Table). The comparison of the clinical and biochemical characteristics between IgAN patients with either RP or SP is summarized in S4 Table.

**Table 2. Base line biochemical characteristics of the CKD cohort and comparison between rapid progressors and stable CKD patients.**

| | Total CKD (n = 514) | Stable patients (n = 255) | Rapid progressors (n = 259) | p-Value |
|---|---|---|---|---|
| Creatinine (umol/L) | 190 (150–242) | 206 (160–262) | 176 (142–223) | <**0.001** |
| eGFR(CKD-EPI) (ml/min/1.73m$^2$) | 28 (19.7–37.5) | 24.7 (18.2–33.1) | 31.4 (23.1–41.4) | <**0.001** |
| Delta eGFR (ml/min/1.73m$^2$/year) | -3 (-4.6 to 0.06) | 0.07 (-0.25 to 0.49) | -4.7 (-6.4 to -3.7) | <**0.001** |
| Urea (mmol/L) | 13.2(10.3–17.4) | 14 (10.7–18.4) | 12.4 (10.1–16.3) | **0.003** |
| Albumin (g/L) | 43 (40–45) | 44 (42–45) | 42 (39–44) | <**0.001** |
| Corrected calcium (mmol/L) | 2.29 (2.2–2.3) | 2.29 (2.2–2.36) | 2.3 (2.2–2.37) | 0.16 |
| Phosphorus (mmol/L) | 1.1 (0.9–1.27) | 1.07 (0.94–1.24) | 1.16 (1.02–1.28) | **0.001** |
| PTH (pmol/L) | 6.8 (4.1–11.2) | 6.7 (3.8–11) | 6.9 (4.46–11.3) | 0.33 |
| Vitamin D[a] (nmol/L) | 35.9(19.9–55) | 38.9 (23.5–60) | 30.6 (17.3–50) | **0.004** |
| CRP (mg/L) | 2.7 (1.2–5.7) | 2.4 (1.1–5.3) | 3.1 (1.3–6) | 0.06 |
| uPCR (mg/mmol) | 34.2 (14–135) | 20 (10–44) | 85 (25–223) | <**0.001** |
| Urate[b](mmol/L) | 0.44(0.38–0.53) | 0.44 (0.38–0.54) | 0.44 (0.38–0.51) | 0.53 |
| Cholesterol(mmol/L) | 4.5 (3.7–5.4) | 4.3 (3.6–5.1) | 4.7 (4–5.6) | <**0.001** |
| Haemoglobin (g/L) | 124 (114–136) | 125 (116–137) | 122 (113.5–134.5) | **0.035** |
| HbA1c[c](mmol/mol) | 40.7 (36–48) | 41 (37–49) | 40 (36–45.4) | 0.09 |

eGFR-estimated glomerular filtration rate calculated using CKD-EPI equation, PTH-parathyroid hormone, CRP- C-reactive protein, uPCR-urine protein:creatinine ratio, HbA1c-haemoglobin A1c.

Variables are expressed as median (interquartile range) and p-Value by Man-Whitney U test.

a- Vitamin D results were only available in142 (55.7%) of slow progressors and 143 (55.2%) of rapid progressors.

b- Urate results were only available in 199 (78%) of slow progressors and 202 (78%) of rapid progressors.

c- HbA1C were only available in 201(78.8%) of slow progressors and 214(82.6%) of rapid progressors.

## Study outcomes

1. CKD progression: median follow-up for the CKD patients was 56(IQR 36–83) months with significantly longer follow-up in SP (median 71; IQR 47–104 months) than RP (median 43; IQR 30–62 months), p < 0.001;238 (91.9%) of the RP and 23 (9%) of the SP patients reached the progression end-point during the study follow-up period.

2. Mortality: median follow-up for the CKD patients was 57(IQR38-84) months, with significantly longer follow-up in SP (median 71; IQR 47–104 months) than RP (median 48; IQR

**Table 3. Comparison of genotype variant and allele frequency between CKD and control group, and between RP and SP patients.**

| | | CKD | | | Controls (n = 454) | p-Value |
|---|---|---|---|---|---|---|
| | Total (n = 514) | Stable patients (n = 255) | Rapid progressors (n = 259) | p-Value | | |
| *Complement 3 (rs2230199)* | | | | | | |
| *FF* | 48 (9.3%) | 19 (7.5%) | 29 (11.2%) | | 26 (5.7%) | |
| *FS* | 168 (32.7%) | 86 (33.7%) | 82 (31.7%) | 0.34 | 135 (29.7%) | **0.039** |
| *SS* | 298 (58%) | 150 (58.8%) | 148 (57.1%) | | 293 (64.6%) | |
| *Allele frequency* | | | | | | |
| *F* | 264 (25.7%) | 124 (24.3%) | 140 (27%) | 0.32 | 187(20.6%) | **0.008** |
| *S* | 764 (74.3%) | 386 (75.7%) | 378 (73%) | | 721(79.4%) | |

p-Value by Chi-square test.

FF-homozygous complement 3 fast, FS-heterozygous complement 3, SS-homozygous complement 3 slow.

32.5–64.5 months), p < 0.001;36 (13.9%) of the RP and 58 (22.7%) of the SP had died during the study follow-up period.

**C3 polymorphism and CKD progression.** There was no significant association between *C3* polymorphism and CKD progression by Cox regression analysis, hazard ratio (HR) = 1.1 (95% confidence interval (CI) 0.75–1.6; p = 0.59)for *C3FF*, HR = 1.0(CI 0.8–1.3; p = 0.78) for C3FS and HR = 0.95(CI 0.74–1.2; p = 0.70) for *C3SS*. The factors which showed significant association with progression in our CKD cohort in multivariate Cox regression analysis were age, gender, smoking, haemoglobin, baseline eGFR, and urinary protein:creatinine ratio (S5 Table).

**C3 polymorphism and mortality.** There was a significant association between *C3* homozygous variant (*C3FF*) polymorphism and mortality in univariate regression analysis with HR = 1.8(CI 1.4–3.1; p = 0.037). This association remained significant after adjustment for other risk factors (model 1), but lost significance after adjustment for C-reactive protein (CRP) in model 2 with HR = 1.6 (CI 0.9–2.8; p = 0.14). The factors which showed significant association with mortality in our CKD cohort in the multivariate Cox regression were age, CCF, haemoglobin and CRP (Table 4).

## Study 2 (biopsy-proven GN)

**Baseline characteristics.** The GN group was mainly Caucasian (99.3%), hypertensive (92.2%), predominantly male (72.5%), with a median age of 59.9(IQR 47.9–68.5)years.The median baseline eGFR was 33.5(IQR 21.3–46.6)ml/min/1.73m$^2$/yr and median ΔeGFR was-1.6(IQR -4.2 to 0.06)ml/min/1.73m$^2$/yr. Clinical and biochemical baseline characteristics of this group are summarized in S6 Table.

**Table 4. Cox regression analysis (death) univariate and multivariate models (CKD cohort n = 514, events n = 94).**

| Factor | Univariate model HR (95% CI) | p-Value | Multivariate model 1* HR (95% CI) | p-Value | Multivariate model 2* HR (95% CI) | p-Value |
|---|---|---|---|---|---|---|
| *Complement 3 FF* | 1.8 (1.04–3.1) | **0.037** | 1.9 (1.1–3.4) | **0.033** | 1.6 (0.9–2.8) | 0.14 |
| *Complement 3 FS* | 0.82 (0.52–1.3) | 0.41 | | | | |
| *Complement 3SS* | 0.9 (0.59–1.4) | 0.62 | | | | |
| Age | 1.1 (1.06–1.1) | **<0.001** | 1.1 (1.05–1.09) | **<0.001** | 1.1 (1.04–1.09) | **<0.001** |
| Gender (female) | 0.84 (0.54–1.3) | 0.43 | | | | |
| Smoking | 1.68 (1.06–2.6) | **0.027** | 1.5 (0.91–2.4) | 0.12 | 1.4 (0.85–2.3) | 0.18 |
| HTN | 1.2 (0.4–3.7) | 0.81 | | | | |
| DM | 2.0 (1.35–3.07) | **0.001** | 1.1 (0.71–1.7) | 0.68 | 1.1 (0.71–1.7) | 0.70 |
| Tumor | 1.8 (1.05–3.0) | **0.034** | 1.5 (0.86–2.7) | 0.15 | 1.5 (0.85–2.6) | 0.17 |
| MI | 2.6 (1.5–4.3) | **<0.001** | 1.4 (0.83–2.5) | 0.19 | 1.4 (0.83–2.5) | 0.19 |
| CCF | 4.0 (2.1–7.7) | **<0.001** | 2.5 (1.3–5.0) | **0.009** | 2.5 (1.3–5.1) | **0.008** |
| eGFR (CKD-EPI) | 0.97 (0.95–0.98) | **0.001** | 0.99 (0.97–1.0) | 0.52 | 0.99 (0.97–1.0) | 0.32 |
| Albumin (g/L) | 0.96 (0.92–1.0) | 0.05 | | | | |
| Haemoglobin (g/L) | 0.97 (0.96–0.99) | **<0.001** | 0.97 (0.96–0.99) | **0.002** | 0.98 (0.96–0.99) | **0.005** |
| UPCR (g/mol) | 0.99 (0.97–1.0) | 0.17 | | | | |
| CRP (mg/L) | 1.07 (1.04–1.1) | **<0.001** | | | 1.1 (1.02–1.09) | **<0.001** |

*Multivariate model 1 included variables that showed significant association in univariate model except CRP, model 2 after addition of CRP to model 1.FF-homozygous complement 3 fast, FS-heterozygous complement 3,SS homozygous complement 3 slow, HTN-hypertension, DM-diabetes mellitus, MI-myocardial infarction, CCF-congestive cardiac failure, eGFR-estimated glomerular filtration rate calculated using CKD-EPI equation, CRP-C-reactive protein

**Table 5. Genotype variant and allele frequency of the different GN groups.**

| | Total GN (n = 269) | IgA nephropathy (n = 114) | FSGS (n = 50) | Membranous nephropathy (n = 59) | Other GN (n = 46) |
|---|---|---|---|---|---|
| *Complement 3 (rs2230199)* | | | | | |
| **FF** | 23(8.6%) | 9 (7.9%) | 5 (10%) | 5 (8.5%) | 4 (8.7%) |
| **FS** | 80(29.7%) | 40 (35%) | 16 (32%) | 9 (15.3%) | 15 (32.6%) |
| **SS** | 166(61.7%) | 65 (57%) | 29 (58%) | 45 (76.3%) | 27 (58.7%) |
| *Allele frequency* | | | | | |
| **F** | 126(23.4%) | 58 (25.5%) | 26 (26%) | 19 (16%) | 23 (25%) |
| **S** | 412(76.6%) | 170 (74.5%) | 74 (74%) | 99 (84%) | 69 (75%) |

FF-homozygous complement 3 fast, FS-heterozygous complement 3, SS-homozygous complement 3 slow

**The *R102G C3* polymorphism (*rs2230199*).** The genotype variants were (*FF* 8.6%, *FS* 29.7%, *SS* 61.7%) and allele frequencies were (*F* 23.4%, *S* 76.6%) in total GN. The highest C3F frequency was in FSGS (26%) and IgAN (25.5%) and the lowest in membranous nephropathy (16%), (Table 5).

**C3 polymorphism and progression in IgAN patients.** IgAN group was the only GN group to show an association between *C3* polymorphism and progression. 59 (51.8%) of the IgAN patients reached the progression endpoint during the study follow-up period, with median follow-up of 44 (IQR 25–82) months. Cox regression showed a significant association between *C3* polymorphism and CKD progression in the IgAN group with HR = 1.9 (95% CI 1.1–3.1; p = 0.018) for heterozygous *C3FS*, increasing further for homozygous *C3FF* to HR = 2.8 (95% CI 1.2–6.2; p = 0.014). *C3SS* showed a protective benefit for progression with HR = 0.41 (95% CI 0.24–0.68; p = 0.001).These associations remained significant after adjustment for several progression risk factors including treatment with immunosuppressive therapy (Table 6).In a Kaplan–Meier analysis, the incidence rate of renal outcomes was significantly

**Table 6. Coxregression analysis (progression) univariate and multivariate models in IgA nephropathy group (patients n = 114, events n = 59).**

| Factor | Univariate model HR (95% CI) | p-Value | Multivariate Model 1 HR (95% CI) | p-Value | Multivariate Model 2 HR (95% CI) | p-Value |
|---|---|---|---|---|---|---|
| *Complement 3 FF* | 2.8 (1.2–6.2) | **0.014** | 7.8 (3.0–20.2) | **<0.001** | | |
| *Complement 3 FS* | 1.9 (1.1–3.1) | **0.018** | 3.5 (1.9–6.4) | **<0.001** | | |
| *Complement 3 SS* | 0.41 (0.24–0.68) | **0.001** | | | 0.25 (0.14–0.45) | **<0.001** |
| **Age** | 0.99 (0.97–1.0) | 0.15 | | | | |
| **Gender (female)** | 0.99 (0.49–1.9) | 0.95 | | | | |
| **Smoking** | 0.96 (0.57–1.6) | 0.86 | | | | |
| **HTN** | 1.1 (0.3–4.4) | 0.93 | | | | |
| **DM** | 1.2 (0.51–2.8) | 0.69 | | | | |
| **eGFR (CKD-EPI)** | 0.94 (0.92–0.96) | **<0.001** | 0.94 (0.92–0.97) | **<0.001** | 0.94 (0.92–0.97) | **<0.001** |
| **Albumin (g/L)** | 0.91 (0.85–0.97) | **0.005** | 1.1 (9.7–1.2) | 0.16 | 1.05 (0.96–1.1) | 0.27 |
| **Haemoglobin (g/L)** | 0.97 (0.95–0.98) | **<0.001** | 0.97 (0.95–0.99) | **0.005** | 0.98 (0.95–0.99) | **0.011** |
| **UPCR (g/mol)** | 1.02 (1.01–1.02) | **0.001** | 1.02 (1.00–1.03) | **0.009** | 1.01 (1.00–1.03) | **0.027** |
| **CRP (mg/L)** | 1.01 (0.97–1.05) | 0.63 | | | | |
| **Treatment*** | 0.58 (0.36–0.95) | **0.031** | 0.84 (0.49–1.4) | 0.53 | 0.84 (0.49–1.4) | 0.53 |

Multivariate model 1 included the variables that showed significance in univariate model with C3FF/FS. Multivariate model 2 included the variables that showed significance in univariate model with C3SS.

FF-homozygous complement 3 fast, FS-heterozygous complement 3, HTN-hypertension, DM-diabetes mellitus, eGFR-estimated glomerular filtration rate calculated using CKD-EPI equation, UPCR-urine protein:creatinine ratio, CRP- C-reactive protein.

.*Treatment with corticosteroids and or immunosuppressive therapy.

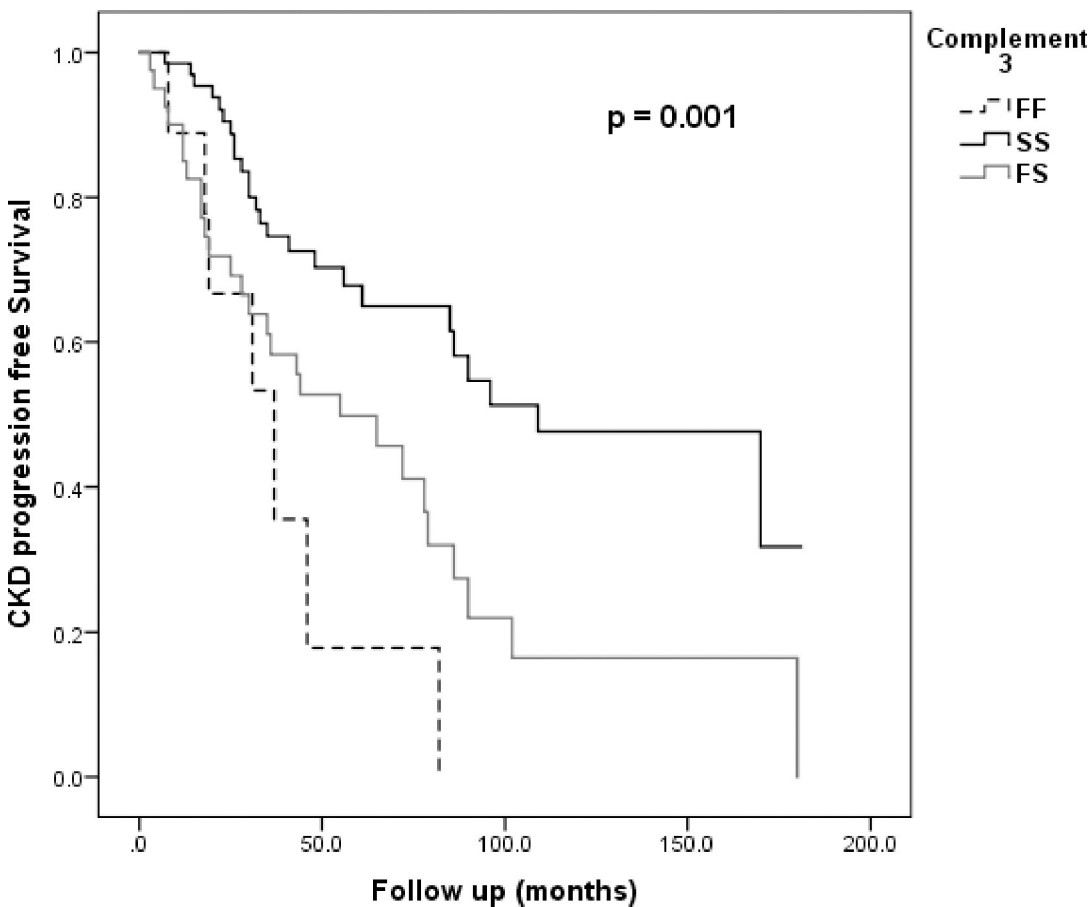

**Fig 3. Kaplan–Meier survival curve for progression in IgAN.** (Patients n = 114, event n = 59); Log Rank 14.8, p-value 0.001.

higher in IgAN patients with *C3FF* and *C3FS* genotypes compared with patients with the *C3SS* genotype (Fig 3).

**C3 deposition in the renal biopsies of the GN patients.** Information regarding C3 deposition assessed by immunoflourescence or immunoperoxidase in the non-sclerosed glomeruli of the renal biopsies was available in 180 of the 269 GN patients. In the all-cause GN group, C3 deposition was positive in 71% of patients with the *C3FF* genotype, 75% of those with *C3FS* genotype and 55% of those with the *C3SS* genotype. In the subgroup analysis by GN type only the IgAN patients (n = 82) showed significant difference in the C3 deposition between patients with different genotypes: 100% of patients with the *C3FF* genotype, 97% of those with *C3FS* genotype and 64% of those with the *C3SS* genotype had C3 deposition in their biopsies, p = 0.002 (Table 7).

## Discussion

Our study has provided insights into the association of the *C3 variant R102G* with CKD progression and mortality in a large non-dialysis CKD cohort.

We observed that the *C3F* allele had a significantly increased frequency in CKD patients than the normal healthy controls. This observation replicates the finding of a previous study undertaken in a Caucasian cohort from Madrid[19]. However, that study reported a *C3F* allele frequency of 40%, much higher than that observed in our CKD cohort (25.7%). Nevertheless,

**Table 7. *C3* deposition in biopsies of the GN patients.**

| GN type | C3 deposition | *C3FF* | *C3FS* | *C3SS* | p-Value |
|---|---|---|---|---|---|
| IgAN (n = 82) | | 6 | 31 | 45 | |
| | Yes | 6 (100%) | 30 (97%) | 29 (64%) | **0.002** |
| | No | 0 (0.00%) | 1 (3%) | 16 (36%) | |
| FSGS (n = 30) | | 4 | 10 | 16 | |
| | Yes | 2 (50%) | 3 (30%) | 6 (37.5%) | 0.88 |
| | No | 2 (50%) | 7 (70%) | 10 (62.5%) | |
| MN (n = 34) | | 1 | 6 | 27 | |
| | Yes | 0 (0.00%) | 6 (100%) | 20 (74%) | 0.89 |
| | No | 1 (100%) | 0 (0.00%) | 7 (26%) | |
| Others (n = 34) | | 3 | 13 | 18 | |
| | Yes | 2 (67%) | 6 (47%) | 3 (17%) | 0.11 |
| | No | 1 (33%) | 7 (53%) | 15 (83%) | |
| Total (n = 180) | | 14 | 60 | 106 | |
| | Yes | 10 (71%) | 45 (75%) | 58 (55%) | **0.03** |
| | No | 4 (29%) | 15 (25%) | 48 (45%) | |

GN glomerulonephritis, IgANimunoglobulin A nephropathy, FSGS focal and segmental glomerulosclerosis, MN membranous nephropathy, C3 complement 3, C3FF fast homozygous complement 3, C3FS heterozygous, C3SS slow homozygous complement 3. Categorical variables are expressed as number (%) and p-Value by Monte-Carlo test.

our results include a larger number of CKD patients compared to the Madrid cohort (514 vs. 20). In our healthy controls the *C3F* frequency was 20.7% which is similar to reports in the literature for Caucasian populations[16].

*C3F* allele frequencies in CKD patients with rapid or slow CKD progression have not been reported in previous studies. We found that the *C3F* allele was more common in RP (27%) than in SP (24.3%) but this difference was not significant. Despite the *C3F* allele frequency being higher in RP than in healthy controls (20.7%), we cannot conclude that an association between *C3F* allele frequency and progression of all cause CKD exists as the difference may simply be due to the fact that *C3F* allele frequency is greater in CKD patients than healthy people. This point was also confirmed by the Cox regression analysis that showed no significant association between *C3F* homozygous or heterozygous status and CKD progression in our CKD cohort.

Most of the GWAS that have been conducted in CKD patients have tested the association between different SNPs and the prevalence of CKD [5–8], There are only 2 studies that have searched for the genetic factors that are associated with progression rather than prevalence of CKD: Boger CA et al [34] and Parsa A et al [35].Boger et al's study tested only 16 SNPs and found that 11 of them associated with incident CKD. Our targeted SNP, the *R102G*, was not one of the 16 tested SNPs in this candidate gene association study. Parsa et al's GWAS tested a million SNPs in a large CKD cohort with 5 years follow up and found 12 SNPs associated with time to ESRD in black patients and 6 SNPs in white patients; the *R102G SNP* was not one of the latter 6 SNPs. This agrees with our finding that the *R102G* was not associated with progression in all-cause CKD. However in Parsa et al's study the specific causes of CKD in their cohort were not defined and so they could not relate primary renal disease-specific progression to specific SNPs.

To the best of our knowledge our study is the first to explore the association between *C3* polymorphism and mortality. We found that there was a significant association of *C3FF* and mortality in the entire CKD cohort, but this became non-significant after adjustment for the CRP levels. It is well-known that higher CRP level is associated with all-cause mortality in the general population as well as in CKD patients[36,37]. In our study, CKD patients with the

*C3FF* status had a significantly higher level of CRP than those with *C3FS* and *C3SS*. This may explain the significant association between *C3FF* and mortality that was found in univariate Cox regression, but which was lost after adjustment for CRP levels in the multivariate analysis.

In the second study, the IgAN was in keeping with the published literature showed strong link between *C3F* and IgAN[20]. In our group of 114 IgAN patients we found a *C3F* allele frequency of 25.5%, which was higher than in normal healthy controls (20.7%).This finding agrees with the previous studies, except for one study that has been conducted in a group of Chinese IgAN patients in which no difference in *C3F* allele frequency between the IgAN and control group was reported[38]. This can be explained by the rarity of this SNP in the Asian population in general (1%) [16] and points to the difference in the genetic background of the IgAN in different ethnicities.

IgAN is considered the commonest GN worldwide. However, the role of the complement system in the pathogenesis of IgAN is still unclear; mesangial deposits of some complement components including C3, complement factor H (CFH) and complement factor H related protein 5 (CFHR5) are found in the renal biopsies from IgAN patients[39]. Whether galactose-deficient IgA1 containing immune complexes can activate the complement system in IgAN patients while circulating in plasma or after their deposition in the kidney still needs to be elucidated [40]. We found a significant difference between the three *C3* genotypes with regards the deposition of C3 in the renal biopsies of IgAN patients with significantly more deposition in patients with *C3FF* and *C3FS* genotypes than in those with the *C3SS* genotype. This may indicate a difference in the affinity of complement binding to the galactose-deficient IgA1 containing immune complexes in IgAN patients according to their *C3* genotype, but this is speculative and requires further investigation.

The previous study that found increased *C3F* frequency in IgAN patients also reported that this increase was particularly noted in those with renal impairment or hypertension[20]. In subgroup analysis of the first part of our study we found that *C3F* allelefrequency was significantly higher in RP IgAN patients than SP with more than six-fold increased risk. Also, in the Cox regression analysis we found a strong association between *C3F* homozygous and heterozygous status and CKD progression in 114 IgAN patients. We have thus provided novel data highlighting that IgAN patients who are carriers of the *C3F* allele are at increased risk for rapid progression, while the *C3SS* status confers a protective benefit against progression.

Although our study is limited by a sample size, it benefits from a robust methodology used to select the RP and the SP patient status resulting in exclusion of a significant number of potentially confounding data from other patients. Beside the robust methodology our study is also strengthened by the data being derived from an advanced CKD cohort (stages 3–5 not on dialysis) with a long follow-up period (average of 4 years) which therefore included a large number of events that strengthened the determination of genetic associations.

## Conclusion

*C3 SNP (R102G)* is associated with rapid CKD progression in IgAN patients but not in other causes of CKD. Further research is required to replicate the association and fully elucidate the pathophysiological mechanism of this association, which could help unravel novel targets for treatments.

## Supporting information

**S1 Table. Calculation to determine whether observed genotype frequencies are consistent with Hardy-Weinberg equation.**
(DOCX)

**S2 Table. Comparison of baseline characteristics (clinical and biochemical) between CKD patients with the *C3FF*, and those with *C3FS* or *SS*.**
(DOCX)

**S3 Table. Comparison of allele frequency between rapid and stable CKD patients in different causes of CKD.**
(DOCX)

**S4 Table. Comparison of clinical and biochemical characteristics between RP IgAN patients and SP IgAN patients is summarized in S4 Table.**
(DOCX)

**S5 Table. Cox regression analysis (renal progression) univariate and multivariate models.**
(DOCX)

**S6 Table. Baseline characteristics (clinical and biochemical) of the different GN groups.**
(DOCX)

## Acknowledgments

We would like to thank the renal patients and renal staff involved in the Salford Kidney Study for their contribution to this work.

## Author Contributions

**Conceptualization:** Sara T. Ibrahim, Rajkumar Chinnadurai, Ibrahim Ali, Philip A. Kalra.

**Data curation:** Sara T. Ibrahim.

**Formal analysis:** Sara T. Ibrahim, Rajkumar Chinnadurai, Ibrahim Ali, Philip A. Kalra.

**Investigation:** Sara T. Ibrahim, Debbie Payne, Gillian I. Rice.

**Methodology:** Sara T. Ibrahim, Rajkumar Chinnadurai, Ibrahim Ali, Philip A. Kalra.

**Supervision:** William G. Newman, Eman Algohary, Ahmed G. Adam.

**Writing – original draft:** Sara T. Ibrahim, Rajkumar Chinnadurai, Ibrahim Ali, Philip A. Kalra.

**Writing – review & editing:** Sara T. Ibrahim, Rajkumar Chinnadurai, Ibrahim Ali, William G. Newman, Eman Algohary, Ahmed G. Adam, Philip A. Kalra.

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
