## [Decision Letter · Decision Letter 0]

13 Nov 2019

PONE-D-19-25255

Genetic polymorphism in C3 is associated with progression in chronic kidney disease (CKD) patients with IgA nephropathy but not in other causes of CKD

PLOS ONE

Dear Dr Ibrahim,

Thank you for submitting your manuscript to PLOS ONE. After careful consideration, we feel that it has merit but does not fully meet PLOS ONE’s publication criteria as it currently stands. Therefore, we invite you to submit a revised version of the manuscript that addresses the points raised during the review process.

**The manuscript focuses on a topic of potential interest. However, the study has several shortcomings that should be addressed. To mention few of them, i) concern about the novelty regarding the association of the R102G and CKD; ii) need to analyze the results of the study considering the already available GWAS data on CKD; iii) concern about the sample size of the 37 patients with IgA nephropathy too small to be convincing of a different distribution of the polymorphism on the RP and SP groups; iv) need to evaluate the pathophysiological link between the R102G polymorphism and complement activation; v) need to include in the multivariate Cox regression model the evaluation of the IgAN subgroup CRP; vi) concern about the fact that the study featured a cohort that was primarily Caucasian which limits the generalization of the results; vii) concern about the significant differences in age between RP ad SP likely contributing to findings; viii) need to indicate how common the C3F variant is in the general population; ix) concern about the fact that Table 6 shows a regression analysis of the IgAN group in which the multivariate regression analysis did not include the C3SS genotype as a variant**.

We would appreciate receiving your revised manuscript by Dec 28 2019 11:59PM. To enhance the reproducibility of your results, we recommend that if applicable you deposit your laboratory protocols in protocols.io, where a protocol can be assigned its own identifier (DOI) such that it can be cited independently in the future. For instructions see: http://journals.plos.org/plosone/s/submission-guidelines#loc-laboratory-protocols

We look forward to receiving your revised manuscript.

Kind regards,

Giuseppe Remuzzi

Academic Editor

PLOS ONE

Journal Requirements:

2. In the ethics statement in the manuscript and in the online submission form, please provide additional information about the patient records used in your retrospective study. Specifically, please ensure that you have discussed whether all data were fully anonymized before you accessed them and/or whether the IRB or ethics committee waived the requirement for informed consent. If patients provided informed written consent to have data from their medical records used in research, please include this information.

Reviewers' comments:

Reviewer's Responses to Questions

**Comments to the Author**

1. Is the manuscript technically sound, and do the data support the conclusions?

Reviewer #1: Partly

Reviewer #2: Partly

2. Has the statistical analysis been performed appropriately and rigorously? 

Reviewer #1: I Don't Know

Reviewer #2: Yes

3. Have the authors made all data underlying the findings in their manuscript fully available?

Reviewer #1: No

Reviewer #2: Yes

4. Is the manuscript presented in an intelligible fashion and written in standard English?

Reviewer #1: Yes

Reviewer #2: Yes

5. Review Comments to the Author

Reviewer #1: In this paper the authors provide evidence for the association of a genetic variant with CKD in a clinical study. Their goal was to identify novel genetic risk factors that lead to progression of kidney diseases. The target of interest was a variant in the human complement component C3. The study shows that within the cohort, there was a statistical overrepresentation of the C3FF variant in patients with fast progressing CKD and IgAN.

The study featured a cohort that was primarily caucasian which limits the impact of the results.

Significant differences in age between RP and SP likely contributes to findings, given likely strong survival bias. More clarification on how age when adjusted for affects the results.

The data in tables 1, 2, and 3 features multiple observations on various measurements, with a p-value determined by a Mann-Whitney U test or chi-square analysis. In the figure there are 15 observations made, and there should be some sort of adjustment for the false discovery rate as it is misleading to mark observations as significant otherwise.

The paper does not state how common the C3F variant is in the general population, which would affect the interpretation of the observed frequency of the C3FF variant in the patients.

Table 6 shows a regression analysis of the IgAN group in which the multivariate regression analysis did not include the C3SS genotype as a variant, although the p-value for the univariate model seems to be most statistically significant.

Overall, the manuscript presents interesting findings for which the methods are clearly written out and presented. However, the statistical analysis of the findings need further explanation and clarification, with the issues presented above.

Reviewer #2: In this study Sara T Ibrahim and collaborators evaluated the role of the R102G variant in complement 3 (C3) in a coohort of CKD patients. They analyzed the distribution of the polymorphism compared to a healthy control group, in the CKD rapid progressive group (RP) against the CKD stable function subjects (SP). Finally, they evaluated the role of the polymorphism in the sub group of the biopsy proven glomerulonephritis affected patients. C3F allele frequency was found to be significantly higher in the CKD cohort compared with the healthy control group. There was no significant difference in C3F allele frequency between the RP and SP groups. In the glomerulonephritis subgroup Cox regression showed an association between C3F and progression only in those with IgA nephropathy.

Major concerns

The role of complement in many renal conditions and in particular in many glomerulonephrites (IgAN, C3 glomerulonephritis, HUS, membranous nephropathy, MPGN IgG mediated and others) is well known. The evaluation of a C3 polymorphism can be interesting. The authors recognize the small size of their cohort, but justify the importance of their study in consideration of the better clinical characterization of their sample compared to much larger GWAS studies.

However the results of the study are not novel regarding the association of the R102G and CKD: in particular the same evaluation could be better analyzed considering the already available GWAS data on CKD.

The sample size of the 37 patients with IgA nephropathy is too small to be convincing of a different distribution of the polymorphism in the RP and SP groups. Before publication it is necessary to increase the IgAN sample size of a factor of 10 times at least.

Most important the authors did not try to make any evaluation of the pathophysiological link between the R102G polymorphism and complement activation. I understand that the C3F and C3S are two variants with different electrophoretic migration characteristic. What is the difference in C3 activity of the two variants.

Minor concern

In the multivariate Cox regression evaluation of the IgAN subgroup CRP has not been included in the model. Because it was highly colinear with R102G polymorphism in the analysis of the outcome of risk of death it should be maintained even in the other analyses.

6. PLOS authors have the option to publish the peer review history of their article (what does this mean?). If published, this will include your full peer review and any attached files.

Reviewer #1: No

Reviewer #2: Yes: RICCARDO MAGISTRONI

---

## [Author Response · Author response to Decision Letter 0]

27 Nov 2019

Response to reviewers

Dear editor and reviewers,

Thank you so much for reviewing our manuscript and providing us with your comments and queries, aiming to improve our work. The following is our reply to your comments and queries.

1- Concern about the novelty regarding the association of the R102G and CKD

The association between the R102G polymorphism of C3 and CKD or GN is not a novel one and it has been addressed by some studies before. This is the main point we have relied upon in the rationale of our study that the R102G may be associated with progression of all-cause CKD or in specific sub-types of GN (hence study 2 which included all the biopsy proven GN patients in our cohort). The novel point in our study is the association between the R102G and progression in IgA nephropathy patients. This association has not been addressed by any of the previous studies.

-We have now illustrated this point in a clearer way in the introduction and the discussion of the revised manuscript.

2- Need to analyze the results of the study considering the already available GWAS data on CKD

Most available GWAS that have been conducted in CKD [1-5] or IgA nephropathy [6-8] patients aimed to investigate the genetic factors associated with prevalent CKD or IgAN; as these studies lacked the detailed clinical characteristics of the patients and follow up data they were unable to include progression as an end-point. The only 2 GWAS that attempted to search for genetic factors associated with progression rather than prevalence of CKD are Boger CA et al [9] and Parsa A et al [10].Boger et al`s GWAS tested only 16 SNPs and found that 11 of them associated with incident CKD (during a follow up period of 7 years). Our targeted SNP, the R102G, was not one of the 16 tested SNPs in this GWAS. Parsa et al`s GWAS tested a million SNPs in a large CKD cohort with 5 years follow up and found 12 SNPs that were associated with time to ESRD in black patients and 6 in white patients; the R102G SNP was not one of the latter 6 SNPs and this agrees with our finding that the R102G was not associated with progression in all-cause CKD. However in Pars et al`s study the specific causes of CKD in their cohort were not defined and so they could not relate primary renal disease-specific progression to specific SNPs. 

-We have added a paragraph about this in our discussion in the revised manuscript.

3- Concern about the sample size of the 37 patients with IgA nephropathy too small to be convincing of a different distribution of the polymorphism on the RP and SP groups

This is correct and we mentioned this point in our manuscript`s discussion as a limitation. However in study 2 with larger number of IgAN patients (114) we found that C3F is strongly and independently associated with the progression in IgAN patients. We believe that our study could underpin future studies in larger cohorts of IgAN patients which could then replicate this association.

4- Need to evaluate the pathophysiological link between the R102G polymorphism and complement activation

The pathophysiological link between the R102G and complement activation has been evaluated previously by Heurich M et al [11]. They used C3FF and C3SS from plasma of healthy individuals and added them separately to antibody coated sheep RBCs and then added factor B (FB) to them. They found that the samples containing C3SS needed larger amount of FB to lyse the RBCs. Demonstrating that plasma containing C3FF has more complement activity than that containing C3SS. They further investigated this issue and proved that factor H (one of the complement regulator proteins) bound less well to C3FF than C3SS.

-We have added a paragraph about this issue in the introduction of the revised manuscript.

5- Need to include in the multivariate Cox regression model the evaluation of the IgAN subgroup CRP

-We have added the CRP to the univariate Cox regression in table 6 in the revised manuscript. It showed no significant association with progression in IgAN patients so we have not added it to the multivariate Cox model. The variables which showed significant association in the univariate model are the only variables that were used in the multivariate model to be consistent with the statistical rule that one variable can be included in the Cox model for every 8-10 patients with the event of interest (we had 59 events). 

6- Concern about the fact that the study featured a cohort that was primarily Caucasian which limits the generalization of the results

All genetic studies should define the ethnicity of the studied cohort according to the prevalence of studied SNPs. If the studied SNP prevalence is ˂ 5% in a certain ethnicity the presence or the absence of the association between this SNP and the disease will not be accurate due to the rarity of the SNP in this ethnicity. The frequency of the C3F allele differs in different ethnicities: Caucasian (20%), black (5%) and Asian (1%) [12]. Hence the Caucasian cohort appears to be the optimal cohort to be used to investigate the R102G SNP in different diseases.

7- Concern about the significant differences in age between RP and SP likely contributing to findings.

The difference between the age of patients in RP and SP groups was only significant in the whole CKD cohort but it was not significant in the sub group of IgAN patients (37 patients: 21 RP and 16 SP). Also the age was not significantly associated with progression in the IgAN patients in the Cox regression (table 6). The Cox regression analysis for death was undertaken in the whole cohort without separation into SP and RP; adjustment for age in the multivariate analysis was also performed (table 4).

-We have added the table comparing the baseline characteristics of the IgAN subgroup to the supplementary tables in the revised manuscript (S4 table).

8- Need to indicate how common the C3F variant is in the general population

As mentioned above the frequency of the C3F allele differs in different ethnicities (Caucasian (20%), black (5%) and Asian (1%)). We have previously referred to its frequency in Caucasian cohorts and in our healthy control group in the discussion section.

-We have now illustrated this point in the introduction of the revised manuscript.

9- Concern about the fact that Table 6 shows a regression analysis of the IgAN group in which the multivariate regression analysis did not include the C3SS genotype as a variant.

Statistically we cannot include C3SS in the same multivariate model with C3FF and C3FS as they are constant or linearly dependent covariates; C3SS = 1- (C3FF + C3FS).

-We have added a second multivariate model to table 6 in the revised manuscript with C3SS (but without C3FF and C3FS) to show that it is still protective of progression in the multivariate model.

All of the above modifications have been highlighted in yellow in the revised manuscript.

Thank you so much,

Sara T Ibrahim

References

1. Kottgen A, Glazer NL, Dehghan A, Hwang SJ, Katz R, Li M, et al. Multiple loci associated with indices of renal function and chronic kidney disease. Nature Genetics. 2009;41:712-7.

2. Kottgen A, Pattaro C, Boger CA, Fuchsberger C, Olden M, Glazer NL, et al. New loci associated with kidney function and chronic kidney disease. Nature genetics. 2010;42(5):376-84.

3. Pattaro C, Kottgen A, Teumer A, Garnaas M, Boger CA, Fuchsberger C, et al. Genome-wide association and functional follow-up reveals new loci for kidney function. PLoS genetics. 2012;8(3):e1002584.

4. Pattaro C, Teumer A, Gorski M, Chu AY, Li M, Mijatovic V, et al. Genetic associations at 53 loci highlight cell types and biological pathways relevant for kidney function. Nat Commun. 2016;7:10023.

5. Chambers J, Zhang W, Lord G, van der Harst P, Lawlor DA, Sehmi JS, et al: Genetic loci influencing kidney function and chronic kidney disease. Nat Genet. 2010;42:373-5.

6. Gharavi AG, Kiryluk K, Choi M, Li Y, Hou P, Xie J, et al: Genome-wide association study identifies susceptibility loci for IgA nephropathy. Nat Genet. 2011;43:321-7.

7. Yu XQ, Li M, Zhang H, Low HQ, Wei X, Wang JQ, et al: A genome-wide association study in Han Chinese identifies multiple susceptibility loci for IgA nephropathy. Nat Genet. 2011;44:178-82.

8. Kiryluk K, Li Y, Scolari F, Sanna-Cherchi S, Choi M, Verbitsky M, et al: Discovery of new risk loci for IgA nephropathy implicates genes involved in immunity against intestinal pathogens. Nat Genet. 2014;46:1187-96.

9. Böger CA, Gorski M, Li M, Hoffmann MM, Huang C, Yang Q, et al. Association of eGFR-Related Loci Identified by GWAS with Incident CKD and ESRD. PLoS Genet. 2011;7(9):e1002292.

10. Parsa A, Kanetsky PA, Xiao R, Gupta J, Mitra N, Limou S, et al. Genome-Wide Association of CKD Progression: The Chronic Renal Insufficiency Cohort Study. J Am Soc Nephrol. 2017;28(3):923-34.

11. Heurich M, Martínez-Barricarte R, Francis NJ, Roberts DL, Rodríguez de Córdoba S, Morgan BP, et al. Common polymorphisms in C3, factor B, and factor H collaborate to determine systemic complement activity and disease risk. Proc Natl Acad Sci U S A. 2011;108(21):8761–6.

12. Bazyar N, Azarpira N, Khatami RS, Galehdari H. The investigation of allele and genotype frequencies of human C3 (rs2230199). Mol Biol Rep. 2012;39(9):8919–24.

---

## [Decision Letter · Decision Letter 1]

18 Dec 2019

PONE-D-19-25255R1

Genetic polymorphism in C3 is associated with progression in chronic kidney disease (CKD) patients with IgA nephropathy but not in other causes of CKD

PLOS ONE

Dear Dr Ibrahim,

Thank you for submitting your manuscript to PLOS ONE. After careful consideration, we feel that it has merit but does not fully meet PLOS ONE’s publication criteria as it currently stands. Therefore, we invite you to submit a revised version of the manuscript that addresses the points raised during the review process.

The revised manuscript is definitely improved. It remains, however, a minor issue raised by Reviewer #2 that should be easily addressed.

We would appreciate receiving your revised manuscript by Feb 01 2020 11:59PM. To enhance the reproducibility of your results, we recommend that if applicable you deposit your laboratory protocols in protocols.io, where a protocol can be assigned its own identifier (DOI) such that it can be cited independently in the future. For instructions see: http://journals.plos.org/plosone/s/submission-guidelines#loc-laboratory-protocols

We look forward to receiving your revised manuscript.

Kind regards,

Giuseppe Remuzzi

Academic Editor

PLOS ONE

Reviewers' comments:

Reviewer's Responses to Questions

**Comments to the Author**

1. If the authors have adequately addressed your comments raised in a previous round of review and you feel that this manuscript is now acceptable for publication, you may indicate that here to bypass the “Comments to the Author” section, enter your conflict of interest statement in the “Confidential to Editor” section, and submit your "Accept" recommendation.

Reviewer #1: All comments have been addressed

Reviewer #2: (No Response)

2. Is the manuscript technically sound, and do the data support the conclusions?

Reviewer #1: Yes

Reviewer #2: Yes

3. Has the statistical analysis been performed appropriately and rigorously? 

Reviewer #1: Yes

Reviewer #2: Yes

4. Have the authors made all data underlying the findings in their manuscript fully available?

Reviewer #1: Yes

Reviewer #2: Yes

5. Is the manuscript presented in an intelligible fashion and written in standard English?

Reviewer #1: Yes

Reviewer #2: Yes

6. Review Comments to the Author

Reviewer #1: Appreciate the additional explanations in the introduction and discussion sections to address concerns.

Reviewer #2: The authors have answered to the questions I raised. However the Parsa et al. study should not be reported as a GWAS (Genome Wide Association Study) but rather as a Candidate Genes Association Study.

7. PLOS authors have the option to publish the peer review history of their article (what does this mean?). If published, this will include your full peer review and any attached files.

Reviewer #1: No

Reviewer #2: Yes: RICCARDO MAGISTRONI

---

## [Author Response · Author response to Decision Letter 1]

19 Dec 2019

Dear 2nd reviewer,

Thank you so much for your comment (The Parsa et al. study should not be reported as a GWAS (Genome Wide Association Study) but rather as a Candidate Genes Association Study).

You are right, but I think you meant the Böger et al study not the Parsa et al study. Parsa et al study is a large GWAS which tested million SNPs but Böger et al study is the one which tested only 16 SNPs that were identified by a previous GWAS. I have now corrected this and highlighted it in yellow in the revised manuscript.

Thanks

Sara T Ibrahim

---

## [Decision Letter · Decision Letter 2]

8 Jan 2020

Genetic polymorphism in C3 is associated with progression in chronic kidney disease (CKD) patients with IgA nephropathy but not in other causes of CKD

PONE-D-19-25255R2

Dear Dr. Ibrahim,

We are pleased to inform you that your manuscript has been judged scientifically suitable for publication and will be formally accepted for publication once it complies with all outstanding technical requirements.

The re-revised version of the manuscript is definitely improved. The authors have adequately addressed all the reviewers’ comments.

With kind regards,

Giuseppe Remuzzi

Academic Editor

PLOS ONE

Additional Editor Comments (optional):

Reviewers' comments:

Reviewer's Responses to Questions

**Comments to the Author**

1. If the authors have adequately addressed your comments raised in a previous round of review and you feel that this manuscript is now acceptable for publication, you may indicate that here to bypass the “Comments to the Author” section, enter your conflict of interest statement in the “Confidential to Editor” section, and submit your "Accept" recommendation.

Reviewer #1: All comments have been addressed

Reviewer #2: All comments have been addressed

2. Is the manuscript technically sound, and do the data support the conclusions?

Reviewer #1: Yes

Reviewer #2: Yes

3. Has the statistical analysis been performed appropriately and rigorously? 

Reviewer #1: Yes

Reviewer #2: Yes

4. Have the authors made all data underlying the findings in their manuscript fully available?

Reviewer #1: Yes

Reviewer #2: Yes

5. Is the manuscript presented in an intelligible fashion and written in standard English?

Reviewer #1: Yes

Reviewer #2: Yes

6. Review Comments to the Author

Reviewer #1: As stated on prior review of submission, the authors have already addressed my comments. Although there are still some limitations, the authors were responsive.

Reviewer #2: (No Response)

7. PLOS authors have the option to publish the peer review history of their article (what does this mean?). If published, this will include your full peer review and any attached files.

Reviewer #1: No

Reviewer #2: Yes: Riccardo Magistroni

---

## [Editor Report · Acceptance letter]

17 Jan 2020

PONE-D-19-25255R2 

Genetic polymorphism in C3 is associated with progression in chronic kidney disease (CKD) patients with IgA nephropathy but not in other causes of CKD 

Dear Dr. Ibrahim:

I am pleased to inform you that your manuscript has been deemed suitable for publication in PLOS ONE. Congratulations! Your manuscript is now with our production department. 

With kind regards,

on behalf of

Prof. Giuseppe Remuzzi 

Academic Editor

PLOS ONE